# Comparative Transcriptome Analysis Identifies Key Defense Genes and Mechanisms in Mulberry (*Morus alba*) Leaves against Silkworms (*Bombyx mori*)

**DOI:** 10.3390/ijms232113519

**Published:** 2022-11-04

**Authors:** Xuejie Zhang, Xinxin Zhu, Yuqian Zhang, Zhicheng Wu, Shoujin Fan, Luoyan Zhang

**Affiliations:** Key Lab of Plant Stress Research, College of Life Science, Shandong Normal University, No. 88 Wenhuadong Road, Jinan 250014, China

**Keywords:** mulberry (*Morus alba*), silkworm (*Bombyx mori*), comparative transcriptome, herbivore response, jasmonic acid signal

## Abstract

As a consequence of long-term coevolution and natural selection, the leaves of mulberry (*Morus alba*) trees have become the best food source for silkworms (*Bombyx mori*). Nevertheless, the molecular and genomic basis of defense response remains largely unexplored. In the present study, we assessed changes in the transcriptome changes of mulberry in response to silkworm larval feeding at 0, 3, and 6 h. A total of 4709 (up = 2971, down = 1738) and 3009 (up = 1868, down = 1141) unigenes were identified after 3 and 6 h of silkworm infestation, respectively. MapMan enrichment analysis results show structural traits such as leaf surface wax, cell wall thickness and lignification form the first physical barrier to feeding by the silkworms. Cluster analysis revealed six unique temporal patterns of transcriptome changes. We predicted that mulberry promoted rapid changes in signaling and other regulatory processes to deal with mechanical damage, photosynthesis impairment, and other injury caused by herbivores within 3–6 h. LRR-RK coding genes (THE1, FER) was predicted participated in perception of cell wall perturbation in mulberry responding to silkworm feeding. Ca^2+^ signal sensors (*CMLs*), ROS (*OST1*, *SOS3*), RBOHD/F, CDPKs, and ABA were part of the regulatory network after silkworm feeding. Jasmonic acid (JA) signal transduction was predicted to act in silkworm feeding response, 10 JA signaling genes (such as OPR3, JAR1, and JAZ1) and 21 JA synthesis genes (such as *LOX2*, *AOS*, and *ACX1*) were upregulated after silkworm feeding for 3 h. Besides, genes of “alpha-Linolenic acid metabolism” and “phenylpropanoid biosynthesis” were activated in 3 h to reprogram secondary metabolism. Collectively, these findings provided valuable insights into silkworm herbivory-induced regulatory and metabolic processes in mulberry, which might help improve the coevolution of silkworm and mulberry.

## 1. Introduction

Mulberry (*Morus alba*) displays tolerance to many metals, including copper, cadmium, and zinc [1,2,3]. Moreover, mulberry is highly adaptable to abiotic stresses and widely distributed. Mulberry trees are generally planted in warm and moist areas, including southeastern Asia, Europe, and North and South America. As an important food source for silkworms (*Bombyx mori*), mulberry leaves possess various nutrients, such as proteins, soluble sugars, and fat, which play an essential role in the growth and development of silkworms [2,4,5,6]. In addition, mulberry leaves are characterized by palatable, highly digestible, and barrier-free feeding properties. Furthermore, they contain many biologically active substances, exerting beneficial effects on improving immunity, anti-inflammatory, and anti-oxidant activities [5,7]. Although many studies have shown the regulatory mechanism underlying silkworm metabolism and silk protein synthesis, few studies have focused on how mulberry trees respond to silkworm feeding.

Plants are continuously exposed to attacks from different herbivores. Plants have evolved complex defense mechanisms after the million years of selection pressure exerted by insect herbivores [8,9]. Specific recognition of the herbivore is required for an efficient defense response, which is then translated into defense signaling to reprogram cellular functions. Plants recognize herbivory via binding herbivore- and damage-associated molecular patterns (HAMPs and DAMPs) to pattern recognition receptors (PRRs) [10,11]. Many HAMPs have been isolated from insect herbivores and information regarding their interactions with PRRs is emerging. Recently, a leucine-rich repeat receptor kinase (LRR-RK) from rice has been shown to be essential for perception and defense against the striped stemborer (SSB) *Chilo suppressalis* [8,9,10,11]. DAMPs can trigger defense responses. For example, oligogalacturonides are pectic fragments perceived by wall-associated kinases in *Arabidopsis thaliana* [12,13]. Chewing herbivores induce mechanical damage, and such damage markedly changes the extracellular space by liberating cell wall fragments and intracellular components. The injury activates the early signaling steps, including (i) depolarization of the plasma transmembrane potential (Vm), (ii) increase in cytosolic Ca^2+^, (iii) generation of reactive oxygen species (ROS), and (iv) mitogen-activated protein kinase (MAPK) activity [8,11].

PRRs recognize DAMPs from injured cells at the plasma membrane [14]. Membrane depolarization (Vm), Ca^2+^ signaling, ROS signaling, and downstream MAPK signaling are activated, triggering the biosynthesis of Jasmonoyl-l-isoleucine (JA-Ile), the bioactive form of jasmonic acid (JA) [8,11,15]. Chloroplast- and peroxisome-located enzymes are rapidly activated to generate the primary JA burst through unknown mechanisms. The binding of JA-Ile to the SCFCOI1 complex results in the degradation of JAZ repressors [8,16]. Such degradation further activates transcription factors (TFs), which regulate the synthesis of defense metabolites and proteins. For example, H_2_O_2_ accumulation can be stimulated by Ca^2+^-activated NADPH oxidases (RBOHD and RBOHF) or by GOX in oral secretions [17,18]. Previous studies have identified several negative regulators contributing to the finetuning of the JA pathway. The metabolite-related defense mechanisms include direct defenses such as activating a proteinase inhibitor, polyphenol oxidase, and peroxidase; and indirect defenses such as releasing a blend of terpenoids, phenylpropanoid compounds, and volatile fatty acid derivatives [8,11,19].

Plant structural traits such as leaf surface wax, thorns or trichomes, and cell wall thickness/and lignification form the first physical barrier to feeding by the insects; the secondary metabolites act as toxins and also affect growth and development; and digestibility reducers form the next barriers that defend the plant from subsequent attack [20,21]. The induced damage is transmitted to signal pathways, followed by transcriptomic changes and the induction of biosynthetic pathways. Among available high-throughput analyses, the transcriptome is a powerful approach to explore gene expression in response to herbivory, and such an approach is used in various plant species, notably *A. thaliana*, *N. attenuate*, cotton and tea plant [22,23,24,25,26]. These investigations have shown a link between a plant’s response to herbivore infestation and considerably changed gene expression. 

Although many studies have shown the regulatory mechanism underlying silkworm metabolism and silk protein synthesis, few studies have focused on how mulberry trees respond to silkworm feeding. In the present work, we assessed the dynamic transcriptome of mulberry plants in response to silkworm feeding. Moreover, we explored the target genes and critical signaling implicated in activating herbivory-induced indirect defense.

## 2. Results

### 2.1. Transcriptome Profiling of Mulberry Plants

The leaf samples were sequenced for gene expression at 0, 3, 6 h after silkworm feeding (Figure 1A–D). After sequencing with the Illumina HiSeq X platform, a total of 40,747,346 to 46,219,430 pair-end reads were obtained from nine samples. De novo transcriptome assembly generated 38,076 unigenes, with an average length of 1460 nt and N50 of 2390 nt. On average, 79.73% of the reads from twelve samples were mapped to the reference genome (Appendix A).

### 2.2. Functional Annotations of Unigenes 

Similarity searches by BLASTX were performed to annotate unigenes against different databases. All 38,076 (100%) unigenes were annotated in at least one database. A total of 24,632 (64.69%), 23,936 (62.86%), and 17,375 (45.63%) unigenes showed similarity to sequences in NR (NCBI non-redundant protein sequences, https://www.ncbi.nlm.nih.gov/, accessed on 1 June 2021), NT (NCBI non-redundant protein sequences) and PFAM (protein families database, http://pfam-legacy.xfam.org/, accessed on 1 June 2021) databases with an E-value threshold of 1 × 10^−5^ (Appendix A). A total of 17,374 (45.62%) unigenes were annotated in the GO database by Blast2GO v2.5 with an E-value cutoff of 1 × 10^−6^. A total of 32,179 unigenes of the mulberry were assigned to A. thaliana gene IDs for GO annotation mapping by BLASTX with an E-value cutoff of 1 × 10^−5^ and were used for MapMan analysis.

### 2.3. Differentially Expressed Genes (DEGs) Calculation

The relative level of gene expression in mulberry leaves was evaluated by the FPKM values, which were calculated based on the uniquely mapped reads. Totals of 4709 (Up = 2971, Down = 1738), 3009 (Up = 1868, Down = 1141) unigenes were filtered as dysregulated genes in comparison of EL_3h vs. CK, EL_6h vs. CK with the cutoff of padj < 0.05 and |log2(foldchange)| > 1 (Figure 2A,B). Overlapping studies found that there were 1431/746 common up-/downregulated genes for EL_3h vs. CK and EL_6h vs. CK. Samples of EL_3h vs. CK had a relatively large number of characteristics dysregulated genes (Figure 2C,D).

### 2.4. DEGs at Two Time Points and Function Enrichments

To further provide insights into the functional transitions towards insect response in mulberry, we clustered the 5708 DEGs into eight clusters using the Euclidean distance clustering algorithm (Figure 3A, Appendix A). The gene ontology (GO) annotation was performed to assign genes to functional categories for each cluster (Figure 3A). Genes belonging to cluster 2 (C2) were mainly expressed at 3 and 6 h under the insect stress, and genes in cluster 6 (C6) were synchronously downregulated from 3 to 6 h under insect stress. The early stage (3 to 6 h) was best represented by 2348 expressed genes in C5 (Figure 3A). This cluster contained a set of genes related to “response to wounding”, “oxylipin biosynthetic process”, “jasmonic acid biosynthetic process” and “response to herbivore” (Figure 3B). Genes included in cluster 6 (C6) were downregulated at 3–6 h and participated in “chloroplast organization”, “photosynthesis, light harvesting in photosystem I”, “chlorophyll biosynthetic process” and “response to light stimulus” (Figure 3B; Appendix A). The genes in C3 (651 genes) were highly expressed in EL_3h samples and represented by genes related to “ribosomal large subunit assembly”, “pectin catabolic process”, and “calcium-mediated signaling” (Figure 3B).

### 2.5. Expression Patterns of Genes of Biotic Attack Responding Pathways

We analyzed the expression patterns of genes of biotic attack responding pathways by MapMan analysis. A total of 842 pathways were mapped by MapMan for these genes, of which, 105 pathways were filtered to be enriched by the dysregulated genes with the cutoff *p*-value < 0.05 (Appendix A). The biotic attack responding pathways enrichment result of MapMan analysis is shown in Figure 4. A total of 206, 98, and 81 DEGs were enriched in “signalling”, “secondary metabolism”, and “cell wall” pathways, most of these genes are upregulated. Totals of 7 up-/2 downregulated genes of “sugar and nutrient physiology”, 4 up-/1 downregulated genes of “Catharanthus roseus-like RLK1” and 24 up-/14 downregulated genes of “calcium” were enriched in the “signalling” pathway (Figure 5A). The DEGs of “secondary metabolism” pathway participate in “isoprenoids mevalonate pathway” (up = 6, down = 0), “phenylpropanoids” (up = 19, down = 5) and “flavonoids” (up = 17, down = 5) (Figure 5B). The “cell wall” pathway included genes involving in “precursor synthesis” (up = 9, down = 0), “cellulose synthesis” (up = 8, down = 0) and “cell wall proteins” (up = 13, down = 1) (Figure 5C). Additionally, “protein synthesis”, “hormone metabolism jasmonate” and “RNA regulation of transcription AP2/EREBP” were also enriched by the DEGs. 

We further analyzed the expression patterns of genes of secondary metabolite pathways that are known to be affected by herbivory, specifically alpha-Linolenic acid, phenylpropanoid, and terpenoid metabolism (Appendix A; Appendix A). Many transcripts from these pathways were differentially expressed upon silkworm feeding. The most enriched pathways were “alpha-Linolenic acid metabolism” (21 upregulated genes), “phenylpropanoid biosynthesis” (25 upregulated genes), followed by “sesquiterpenoid and triterpenoid biosynthesis” and “mevalonate pathway”.

### 2.6. Real-Time Quantitative PCR Validation

To verify the RNA-Seq results, an alternative strategy was selected for the dysregulated unigenes. To verify the RNA-seq results, six over- and six under-expressed unigenes were selected for validation. Primers were designed to span exon–exon junctions (Appendix A). In most cases, the gene expression trends were similar between these two methods, the correlation between the two sets of data was R^2^ = 0.669, the result was shown in Appendix A. For example, the homolog of jasmonate-ZIM-domain protein, Cluster-8107.14010, which was detected by RNA-Seq as an upregulated unigene in the insect feeding samples (Log2 fold change (L_2_fc) of EL_3h v CK, EL_6h v CK = 3.977, 3.320), was also detected significantly overexpressed by qRT-PCR method (Appendix A).

## 3. Discussion

As a silk-producing insect with tremendous economic value, the silkworm is usually fed by mulberry leaves, which provide all the necessary nutrients and water. Such a feeding chain results from the long-term coevolution and natural selection between silkworms and mulberry trees. Although many studies have shown the regulatory mechanisms of silkworm metabolism and silk protein synthesis [2,5,6], few studies have focused on how mulberry trees respond to silkworm feeding. In the present study, we found that silkworm infestation significantly changed the transcriptome of the mulberry leaf at 3 and 6 h. Furthermore, the expressions of most DEGs were rapidly and transiently increased within 6 h upon onset. Accumulating evidence shows that insect feeding can significantly change the host plant transcriptome, including rapid changes in signaling and other regulatory mechanisms and the considerable reprogramming of primary and secondary metabolisms.

The damage caused by herbivorous insects to plants mainly includes mechanical damage, photosynthesis and other internal metabolic pathway impairment. Short time (3–6 h) insect feeding causes damage to the leaves of trees, and the response of trees can be quickly started within 6 h. For example, Wang et al. reported profound transcriptomic changes in tea plants (*Camellia sinensis*). In the present study, DEGs in cluster C6 were remarkably downregulated at 3–6 h (Figure 3A), and functional analysis indicated that short-term insect feeding affected chloroplast organization, chlorophyll biosynthetic process, and light harvesting in photosystem I in mulberry, supporting the findings in the previous study that short-term (within 6 h) chewing by herbivores causes mechanical damage in plants. In the present study, cluster C2 contained genes mainly expressed at 3–6 h, and these genes participate in “response to wounding”, “jasmonic acid biosynthetic process”, and “response to herbivore” (Figure 3B). We predicted the dynamic responses of these genes to silkworm chewing in mulberry. Mulberry promoted rapid changes in signaling and other regulatory processes to deal with mechanical damage, photosynthesis impairment, and other injury caused by herbivores at 3–6 h. 

MapMan enrichment analysis results show that, in mulberry, structural traits such as leaf surface wax, cell wall thickness and lignification form the first physical barrier to feeding by the silkworms. In plants, the physical properties of the wax layer as well as its chemical composition are important factors of preformed resistance. In the present study, three wax metabolism genes were upregulated after 3 h of silkworm feeding, including wax synthase family protein coding gene (Cluster-4656.0, L_2_fc = 4.483). In mulberry, the overexpression of cell wall precursor synthesis, cellulose and lignin genes after silkworm feeding (Figure 5B,C), indicated the biosynthesis of lignin and cellulose and accelerated covalent linkage between lignin and other cell wall polymers, which is linked to cell wall strengthening after silkworm feeding.

Plants possess defense mechanisms to deal with insect herbivores. Specific recognition of the herbivore is a requirement for efficient defense response, which is subsequently translated to defense signaling to reprogram cellular functions. Plants recognize herbivory via the binding HAMPs and DAMPs to PRRs. Many HAMPs have been purified from insect herbivores [8,10,11]. A leucine-rich repeat receptor kinase (LRR-RK) from rice plays an essential role in perception and defense against the striped stemborer (SSB) *Chilo suppressalis* [27]. In the present study, two LRR-RK coding genes (Cluster-8107.12313, L_2_fc = 1.466; Cluster-8107.7028, L_2_fc = 2.315) were upregulated after 3 h of silkworm feeding, and one LRR-RK coding gene (Cluster-8107.7426, L_2_fc = 2.368) was significantly overexpressed after 6 h. Chewing herbivores cause mechanical damage, and such damage significantly alters the extracellular space by releasing cell wall fragments and intracellular components. THESEUS1 is a receptor kinase in plants that senses cellulose-related cell wall integrity [28]. Similarly, *FERONIA* can monitor cell wall integrity in response to salt stress by binding to pectin [29,30]. In mulberry, *THE1* (Cluster-6414.0, L_2_fc = 2.731) and *FER* (Cluster-8107.18497, L_2_fc = 2.476) homologous genes were upregulated after 3 h of silkworm infestation (Figure 5A, Table 1). Therefore, we hypothesized that perception of cell wall perturbation plays a key role in mulberry responding to silkworm feeding.

Upon insect perception, early signaling is reflected by increased cytosolic Ca^2+^, ROS production, and MAPK activity. Different Ca^2+^ sensors can regulate the decoding of Ca^2+^ signals, such as calmodulins (CaMs), calmodulin-like proteins (CMLs), and calcium-dependent protein kinases (CDPKs) [31]. We showed that CML homologous gene (Cluster-8107.14454, L_2_fc = 2.096) was overexpressed in mulberry after 3 h of silkworm feeding (Figure 5A, Table 1). In Arabidopsis, calcium-binding EF-hand family protein SOS3 encodes a calcium sensor fundamental for K^+^/Na^+^ selectivity and salt tolerance [32]. Calcium-independent ABA-activated protein kinase OST1 functions in the interval between ABA perception and ROS production in the ABA signaling network. In addition, wounding results in a rapid local and systemic ROS burst depending on respiratory burst oxidase homologs protein D (RBOHD) in Arabidopsis [17,33]. The over-expression of homologous genes of *SOS3* (Cluster-8107.8609, L_2_fc = 1.700), *OST1* (Cluster-8107.16572, L_2_fc = 2.942), *RBOHD* (Cluster-8107.23569, L_2_fc = 1.905), and *RBOHF* (Cluster-8107.2579, L_2_fc = 3.997) suggested that Ca^2+^ signal sensors, ROS, CDPKs, and ABA were part of the regulatory network after silkworm feeding in mulberry (Figure 5A, Table 1). Since ROS-dependent Ca^2+^ influx and CDPKs can activate RBOHs, they were likely to react closely with Ca^2+^ in mulberry early defense signaling. Herbivory and wounding rapidly activate MAPKs in the plant’s immune system. In mulberry, we found that the homologous *MPK20* gene (Cluster-8107.12324, L_2_fc = 1.930) was upregulated after 3 h of silkworm feeding, indicating that this gene positively regulated resistance to the silkworm.

The perception and early signaling are linked to broad transcriptional reorganization and defense induction by hormonal signaling networks. JA, derived from α-linolenic acid via one branch of the octadecanoid pathway, is a key mediator of defense responses against chewing insects [19]. In response to herbivores, there is a JA signaling pathway in land plants: acyl-lipid hydrolases release α-linolenic acid from galactolipids in plastid membranes and generate 12-oxo-phytodienoic acid (OPDA), which may release defense hormone precursors from membrane lipids [34]. OPDA undergoes three cycles of β-oxidation to form (+)-7-iso-JA and is conjugated to isoleucine by JA-Ile synthase (JAR1). JA-Ile binds to its receptor in the nucleus, the jasmonate-ZIM-domain (JAZ) [16,35]. The degradation of JAZ abolishes repression of MYC TFs, leading to the expressions of defense genes and resistance against various herbivores [31,35]. In mulberry, 10 JA signaling genes and 21 JA synthesis genes were upregulated after 3 h of silkworm feeding (Figure 6A, Table 1), including *LOX2* (Cluster-8107.13480, L_2_fc = 2.673), *AOS* (Cluster-8107.14242, L_2_fc = 2.871), *ACX1* (Cluster-8107.13500, L_2_fc = 2.330), *OPR3* (Cluster-8107.11076, L_2_fc = 2.332), *JAR1* (Cluster-8107.12593, L_2_fc = 1.751), *JAZ1* (Cluster-8107.11608, L_2_fc = 3.341), *JAZ3* (Cluster-8107.14010, L_2_fc = 3.977), and *JAZ10* (Cluster-8107.2271, L_2_fc = 7.087) (Figure 6A). This finding implied that JA signal transduction might function via a similar mechanism to other plants in mulberry against silkworms.

TFs play a fundamental role in mediating defense signaling pathways. Basic helix–loop–helix (bHLH) MYCs regulate defense against herbivores in Arabidopsis by forming a transcriptional complex. Single and higher-order mutants of *MYC2*, *MYC3*, *MYC4*, and *MYC5* exhibit increased susceptibility to *S. littoralis* and *S. exigua* feeding [36,37]. In the present study, *MYC2* (Cluster-8107.13733, L_2_fc = 1.678), *MYC3* (Cluster-8107.1997, L_2_fc = 6.610), and *MYC4* (Cluster-8107.13830, L_2_fc = 2.550) coding genes were significantly over-expressed after silkworm feeding (Table 1). This finding was consistent with the study of Arabidopsis that *myc2*/*3*/*4* mutants display a similarly altered transcriptome in response to feeding, suggesting that MYCs are the main regulators of defense against chewing herbivores in Arabidopsis and mulberry. There were 17 MYBs identified as overexpressed genes in mulberry against silkworm feeding, including *MYB34* (Cluster-8107.6475, L_2_fc = 1.917), *MYB62* (Cluster-8107.25130, L_2_fc = 4.537), and *MYB111* (Cluster-8107.22166, L_2_fc = 4.2). Members of the MYB gene family have experienced many evolutionary events, such as gene duplication, during plant evolution [38,39]. We speculated that MYCs and MYBs formed a functional regulatory network in mulberry. Besides, other TFs with antiherbivore effects were also identified. We found that 10 WRKY TFs were upregulated after short-term feeding with silkworms, including *WRKY40* (Cluster-8107.20775, L_2_fc = 2.107), *WRKY51* (Cluster-8107.24465, L_2_fc = 2.816), and *WRKY19* (Cluster-8107.19064, L_2_fc = 2.0514) (Table 1). A previous study has shown that JAV1-JAZ8-WRKY51 (JJW) complex regulates JA biosynthesis in response to insect attacks [35]. Although we did not identify the orthologous gene of JAV1 in mulberry, we found that *JAZ8* (Cluster-8107.21141, L_2_fc = 4.593) and *WRKY51* were overexpressed after 3 h of silkworm feeding. We speculated that these genes of mulberry regulated responding with silkworm. 

Various secondary metabolites, including phenylpropanoids, flavonoids, and terpenoids, confer resistance to herbivory or function as communication signals between plants and insects [40]. In the present study, many genes involved in phenylpropanoid (25 of 112) and terpenoid (12 of 48) metabolisms were upregulated at 3 h, indicating that silkworm infestation primarily activated the phenylpropanoid and terpenoid pathways in mulberry. Terpenoids can directly repel herbivores. In cotton, feeding chewing herbivores, including *S. exigua* and *Helicoverpa zea*, leads to the release of complex volatile blends, such as β-myrcene, (E)-β-ocimene, DMNT, and (E)-β-caryophyllene, resulting in the increased foraging efficiency of predators and parasitoids [41,42]. In the present work, we showed that silkworm infestation upregulated many genes at the mRNA level in the mevalonate pathway (MVA), such as *ACAT* (Cluster-8107.10778, L_2_fc = 3.619), *HMGCS* (Cluster-8107.12939, L_2_fc = 2.634), *HMGCR* (Cluster-8107.15919, L_2_fc = 4.3524), *MVK* (Cluster-8107.12465, L_2_fc = 3.146), and *FDPS* (Cluster-8107.19051, L_2_fc = 3.353) (Figure 6B). However, essential genes of the non-mevalonate/MEP pathway were not dysregulated in mulberry damaged by the silkworm. Key genes in the mevalonate pathway were upregulated. In cotton, the mevalonate pathway functions as a critical phytoalexin and provides constitutive and inducible resistance against various pests and diseases [24,42]. We speculated that the MVA pathway, not the MEP pathway, was likely to be the main terpenoid pathway in response to silkworm infestation in mulberry.

It is reasonable to point out the limitation of this study. The temporal expression trend of genes provided clues for the involvement of Ca^2+^, JA, TFs, and secondary metabolisms in response to silkworms. However, several questions remain unanswered. For example, how potential PRRs are linked to these early signaling events and how regulatory steps are modulated. Therefore, comprehending the topology of signaling networks of plants in response to herbivory is necessary.

## 4. Materials and Methods

### 4.1. Plant Materials

Mulberry plants (*M. alba*) wildtype were cultivated at the Key Lab of Plant Stress Research, Shandong Normal University, Shandong Province, China. The mulberry seedlings were maintained in a multipurpose and pest-free greenhouse at a temperature of 25 ± 2 °C with a 14/10-h light/dark photoperiod. In addition, 2-month-old mulberry seedling clones grown in pots were used as the experimental materials. In addition, nine healthy mulberry seedlings with consistent growth were chosen for the experiment.

### 4.2. Insect Materials and Insect Feeding Experiment

The silkworms were reared by mulberry leaves at the same institute under the same conditions. Fourth-instar silkworms were used for the insect feeding experiment. Briefly, the fourth-instar silkworms were subjected to food deprivation for 2 h and placed on new shoots of 2-month-old mulberry seedlings. An insect-rearing cage of 15 cm  ×  10 cm  ×  5 cm was adopted for every plant (Figure 1A,B). Three treatments (0, 3, and 6 h) were applied to the mulberry seedlings, and the leaf specimens were harvested at 0, 3, and 6 h after silkworm feeding (Figure 1A,B). There were three biological replicates in each group. Firstly, specimens from “mulberry seedlings 0 h” were collected and used as controls (CK). Then, fourth-instar silkworms were placed on the remaining mulberry seedlings. After 3 h, leaf specimens were harvested from “mulberry seedlings 3 h” and labeled as EL_3h. After 6 h, samples from “mulberry seedlings 6 h” were harvested and recorded as EL_6h. Collected specimens were immediately snap-frozen in liquid nitrogen and then preserved at −80 °C.

### 4.3. RNA Extraction

Three replicates of mulberry control/insect feeding leaves were collected for RNA purification and transcriptome sequencing. Briefly, 1 g of plant materials was ground in liquid nitrogen, followed by total RNA isolation using the TRIzol Reagent (Invitrogen, Carlsbad, CA, USA). The quality and integrity of purified RNA were evaluated using the RNA Nano 6000 Assay Kit of the Agilent Bioanalyzer 2100 system (Agilent Technologies, Santa Clara, CA, USA) and the NanoDrop 2000 spectrophotometer (Thermo Scientific, Wilmington, NC, USA). 

### 4.4. Illumina Library Construction and Sequencing 

The NEBNext^®^ UltraTM RNA Library Prep Kit from Illumina^®^ (NEB, Ipswich, MA, USA) was adopted to generate sequencing libraries. Index codes were supplemented to attribute sequences of each sample. Briefly, poly-T oligo-attached magnetic beads were employed to purify mRNA from total RNA, followed by synthesis of the first- and second-strands of cDNA. Subsequently, the AMPure XP system (Beckman Coulter, Beverly, MA, USA) was used to purify 150~200-bp cDNA fragments. Finally, the size-selected, adaptor-ligated fragments were isolated and enriched by PCR amplification. The PCR amplicons were subjected to high-throughput sequencing on an Illumina HiSeq X platform (Illumina, San Diego, CA, USA). All genetic data were submitted to the NCBI Sequence Read Archive (SRA) database (https://www.ncbi.nlm.nih.gov/sra, accessed on 25 July 2022), SRA accession: PRJNA862319. 

### 4.5. De Novo Assembly of Transcriptome

Reference sequence libraries for mulberry leaves were generated through RNA sequencing and de novo transcriptome assembly. The RNA sample of every accession was sequenced separately. Next, cDNA library construction and Illumina pair-end 150-bp sequencing (PE150) were conducted at Novogene Co., Ltd. (Beijing, China) (http://www.novogene.com/, accessed on 5 March 2021). Clean reads were acquired by removing adapter-containing reads, ploy-N-containing reads, and low-quality reads. The remaining high-quality reads were used for transcriptome assembly using the Trinity software pipeline with default parameters [43]. De novo assembled unigene sequences were used for BLAST searches and annotation against public databases (NR, NT, Swiss-Prot, Pfam, KOG/COG, Swiss-Prot, KEGG Ortholog database, and Gene Ontology) with an E-value threshold of 1 × 10^−5^.

### 4.6. Calculation of Gene Expression 

A total of 9 independent transcript libraries were created for mulberry control/insect feeding leaves. The expressions of genes were assessed by RSEM [44]. The clean reads were aligned to the de novo assembled transcriptome. Gene expression of leaf specimens was determined by the fragment per kilobase of exon model per million mapped reads (FPKM) method [45]. FPKM values were selected to compare the expression levels between fruit and leaf samples with a cutoff of adjusted *p*-value < 0.05 and |log2(foldchange)| > 1.

### 4.7. Bioinformatics Analysis

The unigenes of mulberry were mapped to *A. thaliana* gene IDs by sequence similarity searching against the genome of *A. thaliana* with an E-value cutoff of 1 × 10^−5^. The differentially expressed genes (DEGs) in mulberry samples were subjected to the GO enrichment analysis using the topGO package of R. The DEGs of mulberry unigene IDs were converted to the Arabidopsis TAIR locus IDs. The software KOBAS was adopted to validate the statistical enrichment of DEGs in KEGG pathways in mulberry [46]. KEGG Orthology (KO) annotations of the upregulated genes of EL_3h vs. CK were performed using InterProScan software in the *A. thaliana* genome database. The obtained KOs were submitted to iPath tools (https://pathways.embl.de/, accessed on 10 June 2021) for the phytoalexin biosynthesis pathway analyses. MapMan (version 3.5.1 R2) was chosen for metabolic pathways enrichment for DEGs in leaf samples.

### 4.8. qRT-PCR Verification

The expression patterns identified by the RNA-seq analysis were confirmed by qRT-PCR. The isolated RNA specimens were subjected to DNaseI digestion and reversely transcribed to cDNA using the PrimeScript RT Reagent Kit with gDNA Eraser (Takara, Dalian, China). Six upregulated genes (Cluster-8107.11923, Cluster-8107.1566, Cluster-8107.2073, Cluster-8107.9620, Cluster-8107.23344, and Cluster-8107.14010) and six downregulated genes (Cluster-8107.14617, Cluster-8107.9559, Cluster-8107.12178, Cluster-8107.13660, Cluster-8107.7599, and Cluster-8107.13035) were randomly chosen for the qRT-PCR assay. Ortholog (Cluster-14363.8722) of *A. thaliana* alpha subunit of the elongation factor-1 complex in mulberry was adopted as the housekeeping gene. Premier 5.0 software was employed to design gene-specific primers (21–24 bp) (Appendix A). qRT-PCR was conducted on an ABI7500 Real-Time PCR System (ABI, USA) using SYBR Green qPCR Master Mix (DBI, Germany). Each experiment was performed in triplicate, and the amplification specificity was estimated using melting curve analysis. Relative expressions of the target genes were calculated using the 2-^(ΔΔCt)^ method.

## 5. Conclusions

In the present study, we assessed changes in the transcriptome changes of mulberry in response to silkworm larval feeding at 0, 3, and 6 h. A total of 4709 and 3009 unigenes were identified after 3 and 6 h of silkworm infestation, respectively. The structural traits such as leaf surface wax, cell wall thickness and lignification were predicted to form the first physical barrier to feeding by the silkworms. We predicted that mulberry promoted rapid changes in signaling and other regulatory processes to deal with mechanical damage, photosynthesis impairment, and other injury caused by herbivores within 3–6 h. Ca^2+^ signal sensors, ROS, RBOHD/F, CDPKs, and ABA were part of the regulatory network after silkworm feeding. Jasmonic acid signal transduction might act in mulberry responding silkworm were upregulated after silkworm feeding for 3 h. Additionally, genes of “alpha-Linolenic acid metabolism” and “phenylpropanoid biosynthesis” were activated in 3 h to reprogram secondary metabolism. Collectively, these findings provided valuable insights into silkworm herbivory-induced regulatory and metabolic processes in mulberry, which might help improve the coevolution of silkworms and mulberry.

## Figures and Tables

**Figure 1 ijms-23-13519-f001:**
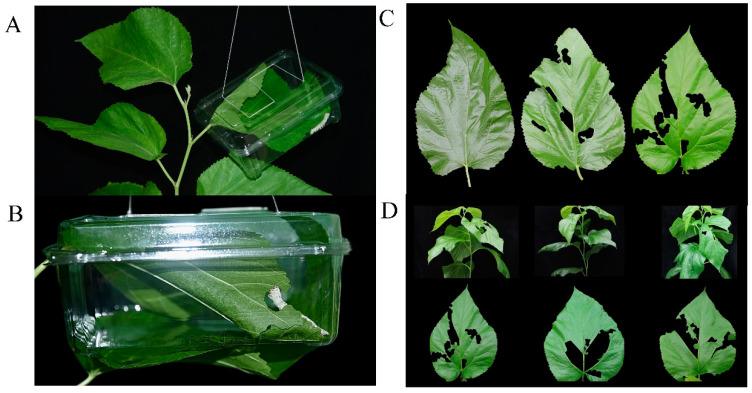
(**A**,**B**) Insect feeding experiment. The fourth-instar silkworms were starved for 2 h and set on the new shoot of 2-month old mulberry seedlings for silkworm feeding treatment. 15 cm  ×  10 cm  ×  5 cm insect rearing cages were used for every separate plant to avoid interference from other pests. (**C**) The phenotype of mulberry leaves after silkworm feeding 0 h (CK), 3 h (EL_3h), and 6 h (EL_6h). (**D**) The phenotype of mulberry seedlings and leaves after silkworm feeding 6 h (EL_6h).

**Figure 2 ijms-23-13519-f002:**
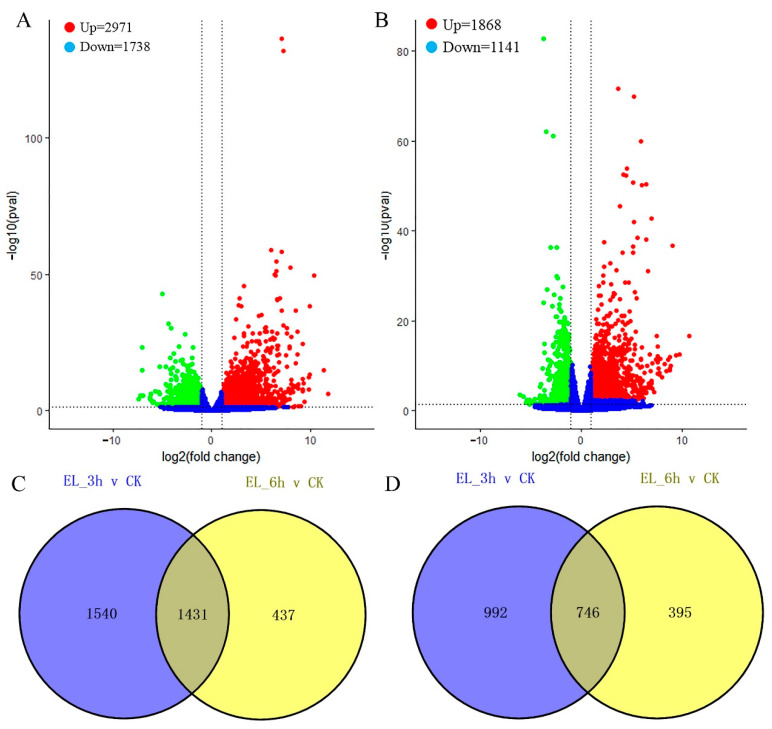
(**A**,**B**) Expression patterns of differentially expressed genes (DEGs) identified between control/insect feeding leaves. Red and green dots represent DEGs, blue dots indicate genes that were not differentially expressed. Totals of 4709 (Up = 2971, Down = 1738), 3009 (Up = 1868, Down = 1141) unigenes were filtered as dysregulated genes in comparison of with the cutoff of padj < 0.05 and |log2(foldchange)| > 1 after 3 and 6 h silkworm infestation. (**C**,**D**) The Venn analysis result of up-/down-regulated genes for 3 and 6 h silkworm infestation samples compared with control samples. There were 1431/746 common up-/downregulated genes.

**Figure 3 ijms-23-13519-f003:**
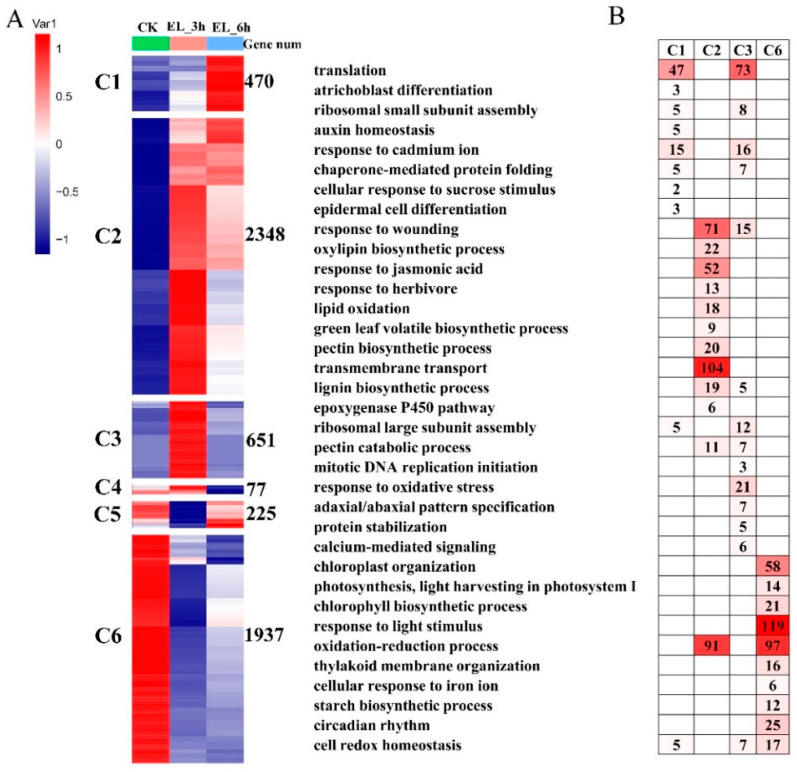
Gene expression pattern and functional transition over the time course. (**A**) Expression patterns of 5708 differentially expressed genes (DEGs) genes in eight DEGs clusters. The number of genes in each cluster are shown on the right. The scale means corrected average expression of genes in each group of samples. (**B**) Gene ontology (GO) enriched in six DEGs clusters. Top significant categories (*p*-value < 0.05) are displayed.

**Figure 4 ijms-23-13519-f004:**
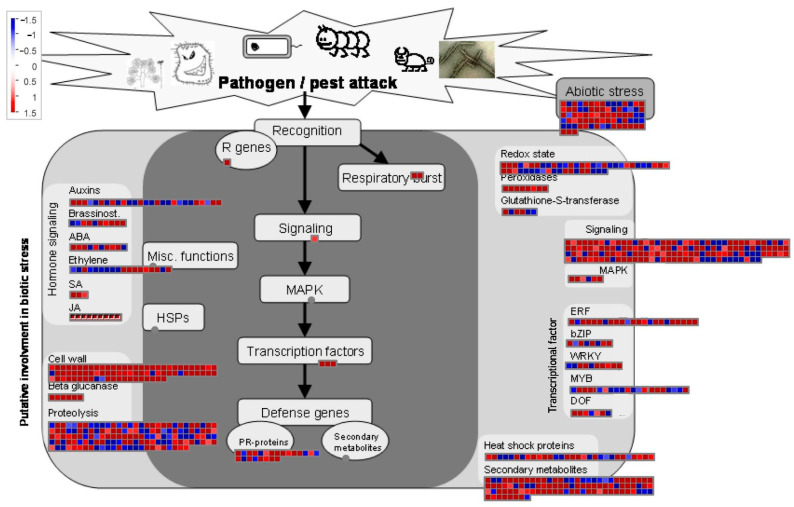
Differently expressed genes (DEGs) viewed globally, which are involved in biotic attack responding pathways. DEGs were chosen for the metabolic pathway analysis using MapMan software. Different colors of boxes indicate the Log2 of the expression ratio of DEGs genes. Totals of 206, 98 and 81 DEGs were enriched in “signalling”, “secondary metabolism”, and “cell wall” pathways, most of these genes are upregulated.

**Figure 5 ijms-23-13519-f005:**
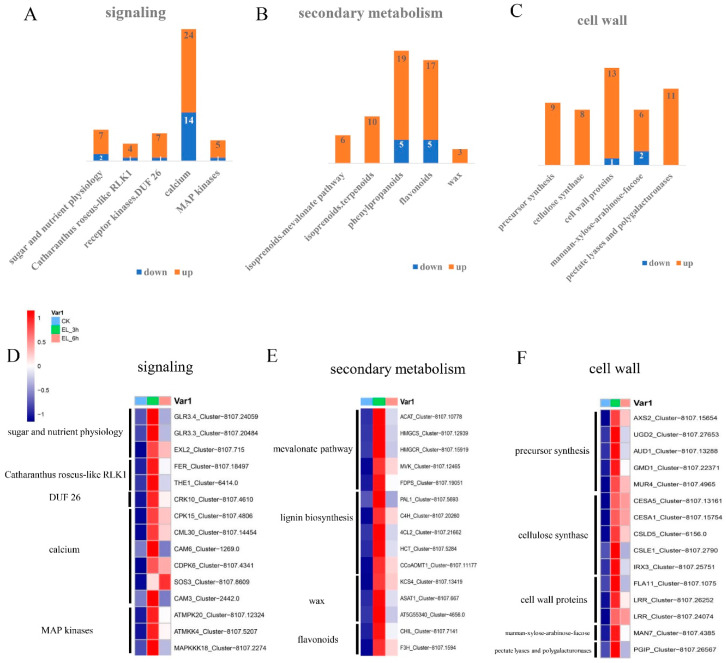
(**A**) DEGs number of secondary processes included in the signaling pathway, the up-/downregulated gene number was marked by orange/blue color. (**B**) DEGs number of secondary processes of the secondary metabolism pathway. (**C**) DEGs number of secondary processes of the cell wall pathway. (**D**) Heatmap of gene expression in control group and silkworm feeding group, the representative functional genes participating in secondary processes of signaling pathway. (**E**) Heatmap of genes in secondary metabolism pathway. (**F**) Heatmap of genes in cell wall pathway.

**Figure 6 ijms-23-13519-f006:**
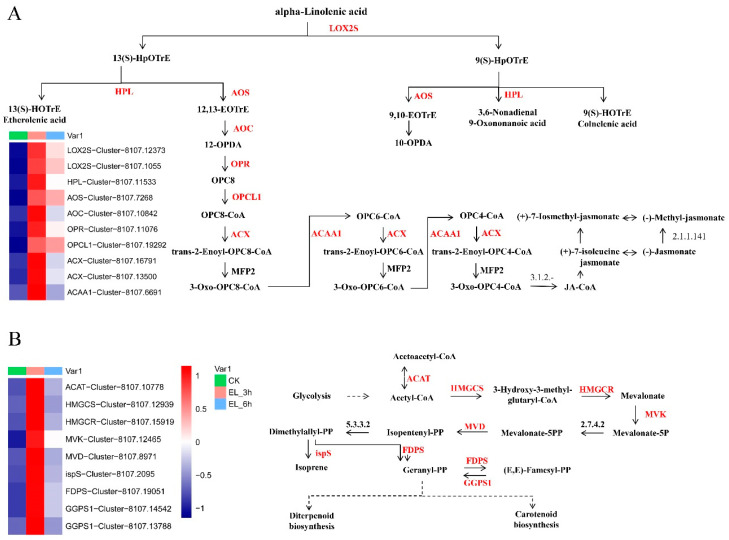
(**A**) Expression patterns of mulberry silkworm-induced genes involved in the alpha-Linolenic acid metabolism pathways. Solid arrows represent established biosynthetic steps. The upregulated elements (genes) are marked in red. LOX2S, lipoxygenase; HPL, hydroperoxide lyase; AOS, hydroperoxide dehydratase; AOC, allene oxide cyclase; OPR, 12-oxophytodienoic acid reductase; OPCL1, OPC-8:0 CoA ligase 1; ACX1, acyl-CoA oxidase; ACAA1, acetyl-CoA acyltransferase. (**B**) Expression patterns of mulberry silkworm-induced genes involved in the terpenoid backbone biosynthesis pathways. ACAT, acetyl-CoA C-acetyltransferase; HMGCS, hydroxymethylglutaryl-CoA synthase; HMGCR, hydroxymethylglutaryl-CoA reductase (NADPH); MVK, mevalonate kinase; MVD, diphosphomevalonate decarboxylase; ispS, isoprene synthase; FDPS, farnesyl diphosphate synthase; GGPS1, geranylgeranyl diphosphate synthase.

**Table 1 ijms-23-13519-t001:** Summary of representative genes participating in silkworm perception and signaling pathways in mulberry.

Process	Gene	Homolog Gene Name	EL_3h v CK L_2_fc	EL_6h v CK L_2_fc
Perception	Cluster-8107.7426	*LRR-RK*	2.099	2.368
	Cluster-8107.12313	*LRR-RK*	1.466	-
	Cluster-8107.7028	*LRR-RK*	2.315	1.318
	Cluster-6414.0	*THE1*	2.731	-
	Cluster-8107.18497	*FER*	2.476	1.555
	Cluster-8107.8882	*RALFL33*	2.274	1.355
Ca^2+^, ROS, MAPK signaling	Cluster-8107.14454	*CML30*	2.096	1.488
	Cluster-8107.8609	*SOS3*	1.700	2.016
	Cluster-8107.24508	*PLP1*	6.244	5.214
	Cluster-8107.4341	*CDPK6*	1.449	1.116
	Cluster-8107.4806	*CPK15*	2.759	2.193
	Cluster-8107.23569	*RBOHD*	1.905	1.409
	Cluster-8107.2579	*RBOH F*	3.997	2.992
	Cluster-8107.12324	*MPK20*	1.93	1.148
	Cluster-8107.2274	*MAPKKK18*	5.565	3.613
JA pathway	Cluster-8107.2271	*JAZ10*	7.087	5.944
	Cluster-8107.14010	*JAZ3*	3.977	3.320
	Cluster-8107.11608	*JAZ1*	3.341	2.389
	Cluster-8107.12593	*JAR1*	1.751	-
	Cluster-8107.12373	*LOX2*	3.154	2.330
	Cluster-8107.7268	*AOS*	3.241	2.330
	Cluster-8107.10842	*AOC*	5.685	4.150
	Cluster-8107.11076	*OPR3*	2.332	1.524
	Cluster-8107.13500	*ACX1*	2.330	1.302
Transcription factors	Cluster-8107.1997	*MYC3*	6.610	6.012
	Cluster-8107.13830	*MYC4*	2.550	2.057
	Cluster-8107.13733	*MYC2*	1.678	1.523
	Cluster-8107.24465	*WRKY51*	2.816	-
	Cluster-8107.19064	*WRKY19*	2.051	1.200
	Cluster-8107.22388	*TTR1*	2.019	-
	Cluster-8107.15321	*WRKY3*	3.348	2.428
	Cluster-8107.20775	*WRKY40*	2.107	-
	Cluster-8107.9774	*MYB73*	2.336	-
	Cluster-8107.1983	*MYB14*	9.886	9.061
	Cluster-8107.1908	*MYB105*	5.440	4.685
	Cluster-8107.25130	*MYB62*	4.537	3.073
	Cluster-8107.27085	*MYB66*	4.582	-
ROS	Cluster-8107.2992	*RAP2.11*	-	2.033
	Cluster-8107.16572	*OST1*	2.942	1.892
	Cluster-8107.18090	*ARGAH2*	5.257	4.760
	Cluster-8107.19259	*UGT73B5*	1.505	-
	Cluster-8107.1416	*ABCG39*	2.261	1.357
	Cluster-8107.19231	*XF1*	1.336	-
	Cluster-8107.12560	*NRAMP2*	1.020	1.079
	Cluster-8107.11277	*APX1*	1.142	-

## Data Availability

All genetic data have been submitted to the NCBI Sequence Read Archive (SRA) database (https://www.ncbi.nlm.nih.gov/sra, accessed on 25 July 2022), SRA accession: PRJNA862319.

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
