# Peer review of "Comparative Transcriptome Analysis Identifies Key Defense Genes and Mechanisms in Mulberry (Morus alba) Leaves against Silkworms (Bombyx mori)"

_ijms, 2022, doi:10.3390/ijms232113519_

Round 1

Reviewer 1 Report

The manuscript presents results of the analysis of transcriptome dynamics in mulberry leaves in response to silkworm larval feeding at early stages (0, 3, and 6 h). The research was performed at the high methodological level, with applying modern methods of functional genomics and bioinformatics. The study highlights the roles of Ca2+, jasmonic acid, transcription factors, some secondary metabolisms (i.e. genes for alpha-Linolenic acid metabolism and phenylpropanoid biosynthesis) in response to silkworms feeding. The topic is relevant, the findings are important for understanding silkworm herbivory-induced regulatory and metabolic processes in mulberry.

 The manuscript is well illustrated, contains six figures, and additional materials.

 At the end of the manuscript the Authors give a critical note on the limitations of the research and consider future perspectives of the development of the studies. This certainly leaves a good impression.

 Comments:

1 The manuscript is generally well written. However, some stylistic corrections are required.

2Figure number is absent on line 170.

3 I have a question concerning the term “susceptible and resistant” mulberry seedlings (line 200). Please give explanation what do you mean. The discussion on this point is absent in the text. 

4 Figure 4 looks too overloaded and illegible. Probably, it will be difficult for visual perception by the readers.

Author Response

Responses to Reviewer 1’s comments:

The manuscript presents results of the analysis of transcriptome dynamics in mulberry leaves in response to silkworm larval feeding at early stages (0, 3, and 6 h). The research was performed at the high methodological level, with applying modern methods of functional genomics and bioinformatics. The study highlights the roles of Ca2+, jasmonic acid, transcription factors, some secondary metabolisms (i.e. genes for alpha-Linolenic acid metabolism and phenylpropanoid biosynthesis) in response to silkworms feeding. The topic is relevant, the findings are important for understanding silkworm herbivory-induced regulatory and metabolic processes in mulberry.

 The manuscript is well illustrated, contains six figures, and additional materials.

 At the end of the manuscript the Authors give a critical note on the limitations of the research and consider future perspectives of the development of the studies. This certainly leaves a good impression.

.

We sincerely thank the reviewer for constructive criticisms and valuable suggestions. For improving the quality of the manuscript.

For figures, we modified Figure1: replaced old panels A, B & C and added new panel C to show the phenotype of mulberry leaves after silkworm feeding 0 h, 3 h and 6 h. We added new Figure 4 to show the expression patterns of genes of biotic attack responding pathways by MapMan analysis. The old Figure 4 was moved to supplementary files as new Figure S2. We deleted old Figure 5, which contained some assumptions that can not be extracted form the obtained data. We added new Figure 5 to show differently expressed genes (DEGs) numbers and heatmaps of secondary processes included in the “signaling pathway”, “secondary metabolism” and “cell wall”.

For tables, we moved the old Table 1 to supplementary files as new Table S1, and added new Table 1 to show summary of representative genes participating in silkworm perception and signaling pathways in mulberry.

For manuscript, we mainly added the results of MapMan analysis and statistical analysis in the results section. In the discussion section, added information about structural traits such as leaf surface wax, cell wall thickness and lignification form the first physical barrier to feeding by the silkworms. In the whole document, we removed all contents about “oligophagous”, “glucosinolate”, “susceptible and resistant”, etc, these content are wrong or there is no more reliable evidence.

The added/modified methods, results and figures highlighted (i) moderate discussion based on gene expression evidence, (ii) show the process of mulberry trees responding to silkworm bites from simple to deep, (iii) focus on signaling pathways and secondary metabolism.

Comments:

1 The manuscript is generally well written. However, some stylistic corrections are required.

We corrected some mistakes in writing, spelling and grammar. For example, silkworm may not really a oligophagous insect. We checked the manuscripts carefully and deleted all the sentences contained “oligophagous”. We modified this sentence.

Line 155-158

“As a silk-producing insect with tremendous economic value, silkworm is usually fed by mulberry leaves, which provide all necessary nutrients and water.”

There is no volatility experiment in this paper. We mistakenly wrote “volatile analysis”. We checked the manuscripts carefully and deleted them.

2Figure number is absent on line 170.

Sorry for the confusion and thanks for suggestion. We added the Figure number.

3 I have a question concerning the term “susceptible and resistant” mulberry seedlings (line 200). Please give explanation what do you mean. The discussion on this point is absent in the text.

Sorry for the confusion and thanks for suggestion. We mistakenly wrote the description of seedlings as “susceptible and resistant”, and the seedlings we used were ordinary wild type. We modified this in the revised manuscript,

  1. 368-369

“Mulberry plants (M. alba) wildtype were cultivated at the Key Lab of Plant Stress Re-search, Shandong Normal University, Shandong Province, China.”

4 Figure 4 looks too overloaded and illegible. Probably, it will be difficult for visual perception by the readers.

Sorry for the confusion and thanks for suggestion. We have increased the clarity of this picture (600dpi) and moved it to supplementary files as new Figure S2. Readers can download it and enlarge it according to their needs.

Reviewer 2 Report

The paper studies the interaction of silkworms on mulberry leaves using a transcriptomics approach. This is quite a specific interaction which has been studied mostly for various aspects of the metabolism and (silk) protein biosynthesis from the insect perspective. Here, the authors aim to address the interaction form the plants’ perspective. The study has chosen a transcriptomics approach which has been technically executed in a solid way. However, the manuscript in its current form is not suitable for publication. See below for my motivation and suggestions.

Abstract:

Line 12 – authors did not describe any measurements of volatile analysis. This is a transcriptome analysis only;. Also check the first line in the conclusion (page 14, line 271).

Line 20 – as it reads now it concerns the silks worms response. Please check this sentence.

Line 23 – as far as i know, mulberry does not produce glucosinolates. See also later comments.

Introduction

Line 32-33 – the second part of this sentence does not relate to the first part. No idea what the authors actually meant here to state.

Line 34 – ‘highly adaptable’  to what?

Line 36 – is silkworm really a oligophagous insect? At least in this sentence it is contractionary: as the sole food source for oligophagous ...

Line 50 – 66 – starts with DAMPS and HAMPS, the latter being specific for herbivory. However, the following text only presents the info on DAMPS. This should be motivated as one would expect info on HAMPS at least in an additional way. However, this is not a plea for extending too much text.

Line 71-73 – i do not understand the relevance of specifically glucosinolate mutants here. I was expecting a more general sentence on the relevance of specialized metabolites in plant-insect interactions.

Line 74-76 – more or less a repetition of above.

Line 77 – phrasing: the transcriptome indicates the collection of transcripts present at a certain point. Here, you most likely mean: transcriptomics analysis

Line 85 – no volatile profiling has been included in this study

Results

Figure 1 – panels A,B & C are not really informative for what happens during the early infestation with silkworms. Better use photo’s like in panel F, to illustrate time point 0, 3 and 6 hrs after introduction of silkworms. In the legend it is named ‘ leafhopper feeding’ – this this another insects to me?

Table 1 – this could move to the supplemental info??

Line 109 – The NR and NT databases are not generally known, please provide full names and references (also for PFAM).

Line 111 – A. thaliana in italics. Please be consistent for this throughout the manuscript.

Figure 2 – here NR and NT are explained but written in small font. Panel A & B are quality assessments, not sure whether this actually contributes to the understanding of the story line. Consider to move to supplemental data). Panel C & D do not have the same scale on the Y-axis, therefore it would be good to increase font size so that the reader have a better understanding. The legend speaks about fast and slow growing cultivars – this is not further presented in the textual description of the data.

Line 135  ...functional transitions along insect response... change to ‘towards’

Line 154 – silkworm (not silkworms)

Figure 5 – indicate what the scale means (fold-change, log fold change, other?) / why were C1, C2, C3, C6 selected in panel B?, why not include all clusters?? The legend speaks of 8 clusters instead of 6.

Line 164 – Shorten sentence: To verify the RNA-seq results, six over- and six under-expressed unigenes were selected for validation .....

Line 170 – indicate which figure

Figure 4, 5 & 6 are not referred to in the results. Figure 4 needs a better explanation of the legend.

In the results section, i do miss some more in depth presentation of the results that are discussed in detail.

In figure 5, there are too many assumptions that can not be extracted form the obtained data. E.g. WRKY’s acting directly on JA-Ile (gene expression, hormone concentration??). Now it is a combination of presenting results (should be in results section) and a hypothesized mode of action (which can indeed be in a discussion figure).

Discussion –

Line 2 – oligophagous, i do not agree as the unique interaction of mulberry and silkworm is considered a monophagous interaction

Line 9 – transient increase can only be stated for the 3-hr timepoint as there is no information on what happens after 6 hrs.

Line 39- 60 while it is very interesting to learn that transcripts of Theseus and Feronia are affected by silkworm feeding, this is not presented in the figures and hence, the proportion of the discussion is a bit out of balance.

Line 66-67 in legend of figure 5 – this is not appropriate to present in the figure legend, but instead should be in the main discussion text. In general legend of this figure is more discussion text, instead of sec description what can be seen in this figure. Text mainly focusses on the response after 3 hrs, and less on those effects after 6 hrs.

Page 10, line 80. While some of the mulberry transcripts show homology with Arabidopsis glucosinolate biosynthetic genes, one cannot claim the presence of glucosinolates (or nicotin, in figure 5) in mulberry. Hereto, a metabolite analysis should be performed. Wat can be stated is that these type of enzymes for which the differential expressed genes encode for (most likely CYP450 enzymes, but this is not clear form the presented data) are involved in specialized metabolism. Then the authors can refer to analogues in Arabidopsis, but be precise in formulation and claiming about specific metabolites present in mulberry.

Figure 6 is a nice figure that shows the fast and transient induction of the a-linolenic and mevalonate pathways in response to silkworm feeding. Please check carefully correct typing (e.g. ios-jasmonate should be isoleucine jasmonate / diteronoid mts be diterpenoid etc) throughout the figure. In panel B it seems that plastidical GGPP (diterpenoids and carotenoids) pathway are mixed with mevalonate FPP pathway. Are the authors aware of compartmentalisation of MVA and MEP pathways? The presented gene transcripts indicate mostly the cytosolic MVA pathway to be affected?

I do not know whether mulberry contains indole diterpene alkaloids as is suggested in figure 6, but if so, the authors should include a reference. Otherwise, it is a similar issue as glucosinolates and nicotine as discussed before.

Line 141 & 149 – GS should be specialized metabolites.

Material and methods

Line 183 – reared on mulberry leaves or other??

Line 200 – susceptible and resistant mulberry seedlings – this has not been addressed in the results and discussion section!

Line 237 – 12 independent libraries – this number does not align with the presented data in this manuscript

Conclusions

Line 271 – volatile (nor endogenous metabolites) analysis is not presented in this study

Line 280 – again the glucosinolates issue... if the authors think this is actually a important issue, i suggest to include a metabolite analysis (or at least a solid reference) to demonstrate the presence of glucosinolates in mulberry.

At this stage of the writing process, i did not check the reference list nor the supplemental data.

Author Response

Responses to Reviewer 2’s comments:

Comments and Suggestions for Authors

The paper studies the interaction of silkworms on mulberry leaves using a transcriptomics approach. This is quite a specific interaction which has been studied mostly for various aspects of the metabolism and (silk) protein biosynthesis from the insect perspective. Here, the authors aim to address the interaction form the plants’ perspective. The study has chosen a transcriptomics approach which has been technically executed in a solid way. However, the manuscript in its current form is not suitable for publication. See below for my motivation and suggestions.

We sincerely thank the reviewer for constructive criticisms and valuable suggestions. For improving the quality of the manuscript.

For figures, we modified Figure1: replaced old panels A, B & C and added new panel C to show the phenotype of mulberry leaves after silkworm feeding 0 h, 3 h and 6 h. We added new Figure 4 to show the expression patterns of genes of biotic attack responding pathways by MapMan analysis. The old Figure 4 was moved to supplementary files as new Figure S2. We deleted old Figure 5, which contained some assumptions that can not be extracted form the obtained data. We added new Figure 5 to show differently expressed genes (DEGs) numbers and heatmaps of secondary processes included in the “signaling pathway”, “secondary metabolism” and “cell wall”.

For tables, we moved the old Table 1 to supplementary files as new Table S1, and added new Table 1 to show summary of representative genes participating in silkworm perception and signaling pathways in mulberry.

For manuscript, we mainly added the results of MapMan analysis and statistical analysis in the results section. In the discussion section, added information about structural traits such as leaf surface wax, cell wall thickness and lignification form the first physical barrier to feeding by the silkworms. In the whole document, we removed all contents about “oligophagous”, “glucosinolate”, “susceptible and resistant”, “volatile”, etc, these contents are wrong or there is no more reliable evidence.

The added/modified methods, results and figures highlighted (i) moderate discussion based on gene expression evidence, (ii) show the process of mulberry trees responding to silkworm bites from simple to deep, (iii) focus on signaling pathways and secondary metabolism.

Abstract:

Line 12 – authors did not describe any measurements of volatile analysis. This is a transcriptome analysis only;. Also check the first line in the conclusion (page 14, line 271).

Sorry for the confusion and thanks for suggestion. There is no volatility experiment in this paper. We mistakenly wrote “volatile analysis”. We checked the manuscripts carefully and deleted them.

Line 20 – as it reads now it concerns the silks worms response. Please check this sentence.

Sorry for the confusion and thanks for suggestion. We modified this sentence.

Line 23 (Line number in the revised manuscript)

“Jasmonic acid (JA) signal transduction was predicted to act in silkworm feeding response,”

Line 23 – as far as i know, mulberry does not produce glucosinolates. See also later comments.

Sorry for the confusion and thanks for suggestion. As suggestion, mulberry does not produce glucosinolates. We checked the manuscripts carefully and deleted all the sentences contained “glucosinolates” or “GS”.

Introduction

Line 32-33 – the second part of this sentence does not relate to the first part. No idea what the authors actually meant here to state.

Sorry for the confusion and thanks for suggestion. We modified this sentence.

Line 34-35

“Mulberry (Morus alba) displays tolerance to many metals, including copper, cadmium, and zinc.”

Line 34 – ‘highly adaptable’  to what?

Sorry for the confusion and thanks for suggestion. We modified this sentence.

Line 35-36

“Moreover, mulberry is highly adaptable to abiotic stresses and widely distributed.”

Line 36 – is silkworm really a oligophagous insect? At least in this sentence it is contractionary: as the sole food source for oligophagous ...

Sorry for the confusion and thanks for suggestion. As suggestion, silkworm may not really a oligophagous insect. We checked the manuscripts carefully and deleted all the sentences contained “oligophagous”. We modified this sentence.

Line 37-39

“As the important food source for silkworm (Bombyx mori), mulberry leaves possess various nutrients, such as proteins, soluble sugars, and fat, which play an essential role in the growth and development of silkworms.”

Line 50 – 66 – starts with DAMPS and HAMPS, the latter being specific for herbivory. However, the following text only presents the info on DAMPS. This should be motivated as one would expect info on HAMPS at least in an additional way. However, this is not a plea for extending too much text.

Sorry for the confusion and thanks for suggestion. We added and modified this.

Line 52-54

“Many HAMPs have been isolated from insect herbivores and information regarding their interactions with PRRs is emerging. Recently, a leucine-rich repeat receptor kinase (LRR-RK) from rice has been shown to be essential for perception and defense against the striped stemborer (SSB) Chilo suppressalis.”

Line 71-73 – i do not understand the relevance of specifically glucosinolate mutants here. I was expecting a more general sentence on the relevance of specialized metabolites in plant-insect interactions.

Sorry for the confusion and thanks for suggestion. As suggestion, mulberry does not produce glucosinolates. We checked the manuscripts carefully and deleted all the sentences contained “glucosinolates” or “GS”.

Line 74-76 – more or less a repetition of above.

Sorry for the confusion and thanks for suggestion. We modified this sentence.

Line 81-82

“The induced damage is transmitted to signal pathways, followed by transcriptomic changes and the induction of biosynthetic pathways.”

Line 77 – phrasing: the transcriptome indicates the collection of transcripts present at a certain point. Here, you most likely mean: transcriptomics analysis

Sorry for the confusion and thanks for suggestion. We modified this sentence.

Line 82-83

“Among available high-throughput analyses, the transcriptome is a powerful approach to explore gene expression in response to herbivory”

Line 85 – no volatile profiling has been included in this study

Sorry for the confusion and thanks for suggestion. There is no volatility experiment in this paper. We mistakenly wrote “volatile analysis”. We checked the manuscripts carefully and deleted them.

Results

Figure 1 – panels A,B & C are not really informative for what happens during the early infestation with silkworms. Better use photo’s like in panel F, to illustrate time point 0, 3 and 6 hrs after introduction of silkworms. In the legend it is named ‘ leafhopper feeding’ – this this another insects to me?

Sorry for the confusion and thanks for suggestion. we modified Figure1: replaced old panels A, B & C and added new panel C to show the phenotype of mulberry leaves after silkworm feeding 0 h, 3 h and 6 h. We mistakenly wrote “leafhopper feeding”, and it was modified as “silkworm” in the revised manuscript.

Line 102-107

“Figure 1. (A-B) Insect feeding experiment. The fourth-instar silkworms were starved for 2 h and set on the new shoot of 2 months old mulberry seedlings for silkworm feeding treatment. 15cm×10cm×5cm insect rearing cages were used for every separate plant to avoid interference from other pests. (C) The phenotype of mulberry leaves after silkworm feeding 0 h (CK), 3 h (EL_3h) and 6 h (EL_6h). (D) The phenotype of mulberry seedlings and leaves after silkworm feeding 6 h (EL_6h).”

Table 1 – this could move to the supplemental info??

Sorry for the confusion and thanks for suggestion. we moved the old Table 1 to supplementary files as new Table S1.

Line 109 – The NR and NT databases are not generally known, please provide full names and references (also for PFAM).

Sorry for the confusion and thanks for suggestion. We provide full names and weblinks for these databases.

Line 112-116

“A total of 24,632 (64.69%), 23,936 (62.86%) and 17,375 (45.63%) unigenes showed similar-ity to sequences in NR (NCBI non-redundant protein sequences, https://www.ncbi.nlm.nih.gov/), NT (NCBI non-redundant protein sequences) and PFAM (protein families database, http://pfam-legacy.xfam.org/) database with an E-value threshold of 1e-5 (Figure S1 A-B).”

Line 111 – A. thaliana in italics. Please be consistent for this throughout the manuscript.

Sorry for the confusion and thanks for suggestion. We checked the manuscripts carefully and modified them.

Figure 2 – here NR and NT are explained but written in small font. Panel A & B are quality assessments, not sure whether this actually contributes to the understanding of the story line. Consider to move to supplemental data). Panel C & D do not have the same scale on the Y-axis, therefore it would be good to increase font size so that the reader have a better understanding. The legend speaks about fast and slow growing cultivars – this is not further presented in the textual description of the data.

Sorry for the confusion and thanks for suggestion. As suggestion, the old Panel A & B were moved to supplementary files as new Figure S1. The old Panel C & D was redrawn and changed to new Panel A & B. We increased the font size in these two pictures, and used the same scale on the Y-axis. We mistakenly wrote “fast and slow growing cultivars”, and we deleted it in the revised manuscript. 

Line 129-136

“Figure 2. (A-B) Expression patterns of differentially expressed genes (DEGs) identified between control/insect feeding leaves. Red and green dots represent DEGs, blue dots indicate genes that were not differentially expressed. Totals of 4709 (Up=2971, Down=1738), 3009 (Up=1868, Down=1141) unigenes were filtered as dys-regulated genes in comparison of with the cutoff of padj<0.05 and |log2(foldchange)| > 1 after 3 and 6 h silkworm infes-tation. (C-D) The Venn analysis result of up-/down-regulated genes for 3 and 6 h silkworm infestation samples compared with control samples. There were 1431/746 common up-/down-regulated genes.”

Line 135  ...functional transitions along insect response... change to ‘towards’

Sorry for the confusion and thanks for suggestion. We modified this sentence.

Line 138-139

“To further provide insights into the functional transitions towards insect response in mulberry, we clustered the 5,708 DEGs into eight clusters using the euclidean distance clustering algorithm”

Line 154 – silkworm (not silkworms)

Sorry for the confusion and thanks for suggestion. We modified this sentence.

Line 178-179

“Many transcripts from these pathways were differentially expressed upon silkworm feeding.”

Figure 5 – indicate what the scale means (fold-change, log fold change, other?) / why were C1, C2, C3, C6 selected in panel B?, why not include all clusters?? The legend speaks of 8 clusters instead of 6.

Sorry for the confusion and thanks for suggestion. The scale means corrected average expression of genes in each group of samples. We added this into the legend. Due to the limitation of the image combination space, only the results of these four clusters are shown in the figure, and the results of other clusters are shown in the supplementary files as new Table S4. The cluster number is 6, we modified this.

Line 155-158

“Figure 3. Gene expression pattern and functional transition over the time course. (A) Ex-pression patterns of 5,708 differentially expressed genes (DEGs) genes in eight DEGs clusters. The number of genes in each cluster are shown on the right. The scale means correct-ed average expression of genes in each group of samples. (B) Gene ontology (GO) enriched in six DEGs clusters. Top significant categories (P_value < 0.05) are displayed”

Line 164 – Shorten sentence: To verify the RNA-seq results, six over- and six under-expressed unigenes were selected for validation .....

Sorry for the confusion and thanks for suggestion. We modified this sentence.

Line 155-158

“To verify the RNA-seq results, six over- and six under-expressed unigenes were selected for validation.”

Line 170 – indicate which figureFigure 4, 5 & 6 are not referred to in the results. Figure 4 needs a better explanation of the legend.

Sorry for the confusion and thanks for suggestion. The old Figure 4 was moved to supplementary files as new Figure S2. We have increased the clarity of this picture (600dpi). Readers can download it and enlarge it according to their needs. We modified the legend of new Figure S2. We checked the reference of each figure throughout the text.

Line 155-158

“Figure S2: Secondary metabolite pathways enriched by up-regulated genes in mulberry after feeding by the silkworm 3 h. The map was generated with iPath (http://pathways.embl.de), a web-based tool for the visualization of metabolic pathways. The enriched pathways were marked by red lines (P_value < 0.05). The most enriched pathways were “alpha-Linolenic acid metabolism” (21 up-regulated genes), “phenylpropanoid biosynthesis” (25 up-regulated genes), followed by “sesquiterpenoid and triterpenoid biosynthesis” and “mevalonate pathway”.”

In the results section, i do miss some more in depth presentation of the results that are discussed in detail.

Sorry for the confusion and thanks for suggestion. For manuscript, we mainly added the results of MapMan analysis and statistical analysis in the results section. In the discussion section, added information about structural traits such as leaf surface wax, cell wall thickness and lignification form the first physical barrier to feeding by the silkworms. The added/modified methods, results and figures highlighted (i) moderate discussion based on gene expression evidence, (ii) show the process of mulberry trees responding to silkworm bites from simple to deep, (iii) focus on signaling pathways and secondary metabolism.

In figure 5, there are too many assumptions that can not be extracted form the obtained data. E.g. WRKY’s acting directly on JA-Ile (gene expression, hormone concentration??). Now it is a combination of presenting results (should be in results section) and a hypothesized mode of action (which can indeed be in a discussion figure).

Sorry for the confusion and thanks for suggestion. We deleted old Figure 5, which contained some assumptions that can not be extracted form the obtained data. We added new Figure 5 to show differently expressed genes (DEGs) numbers and heatmaps of secondary processes included in the “signaling pathway”, “secondary metabolism” and “cell wall”. We added new Table 1 to show summary of representative genes participating in silkworm perception and signaling pathways in mulberry. The added/modified methods, results and figures highlighted (i) moderate discussion based on gene expression evidence, (ii) show the process of mulberry trees responding to silkworm bites from simple to deep, (iii) focus on signaling pathways and secondary metabolism.

Discussion –

Line 2 – oligophagous, i do not agree as the unique interaction of mulberry and silkworm is considered a monophagous interaction

Sorry for the confusion and thanks for suggestion. As suggestion, silkworm may not really a oligophagous insect. We checked the manuscripts carefully and deleted all the sentences contained “oligophagous”. We modified this sentence.

Line 155-158

“As a silk-producing insect with tremendous economic value, silkworm is usually fed by mulberry leaves, which provide all necessary nutrients and water.”

Line 9 – transient increase can only be stated for the 3-hr timepoint as there is no information on what happens after 6 hrs.

Sorry for the confusion and thanks for suggestion. Short time (3-6 hours) insect feeding causes damage to the leaves of trees, and the response of trees can be quickly started within 6 hours. For example, Wang et al. reported profound transcriptomic changes in tea plant (Camellia sinensis). Although there are further response processes such as metabolism and synthesis after 6 hours, since the key response pathways such as cell wall synthesis, signal and secondary metabolism have been started before 6 hours, this study pays more attention to the processes before 6 hours. In addition, a large part of leaves of plants will be lost after 6 hours, which may have a negative impact on the experimental results. We added this into the manuscript.

Line 222-225

“Short time (3-6 hours) insect feeding causes damage to the leaves of trees, and the response of trees can be quickly started within 6 hours. For example, Wang et al. reported profound transcriptomic changes in tea plant (Camellia sinensis).”

while it is very interesting to learn that transcripts of Theseus and Feronia are affected by silkworm feeding, this is not presented in the figures and hence, the proportion of the discussion is a bit out of balance.

Sorry for the confusion and thanks for suggestion. We added the new Table 1 and Figure 5, which focus on signaling pathways. Both of them included Theseus and Feronia.

Line 66-67 in legend of figure 5 – this is not appropriate to present in the figure legend, but instead should be in the main discussion text. In general legend of this figure is more discussion text, instead of sec description what can be seen in this figure. Text mainly focusses on the response after 3 hrs, and less on those effects after 6 hrs.

Sorry for the confusion and thanks for suggestion. In the previous version, some discussion contents were wrongly added to the legend of Figure 5. In the revised version, we deleted the old Figure 5 and modified these contents. The signal pathways and genes included in the old Figure 5 are listed in the new Table 1.

Line 266-285

“Upon insect perception, early signallings are reflected by increased cytosolic Ca2+, ROS production, and MAPK activity. Different Ca2+ sensors can regulate the decoding of Ca2+ signals, such as calmodulins (CaMs), calmodulin-like proteins (CMLs), and calci-um-dependent protein kinases (CDPKs) [31]. We showed that CML homologous gene (Cluster-8107.14454, L2fc= 2.096) was over-expressed in mulberry after 3 h of silkworm feeding (Figure 5A, Table 1). In Arabidopsis, calcium-binding EF-hand family protein SOS3 encodes a calcium sensor fundamental for K+/Na+ selectivity and salt tolerance [32]. Calcium-independent ABA-activated protein kinase OST1 functions in the interval be-tween ABA perception and ROS production in the ABA signaling network. Besides, wounding results in a rapid local and systemic ROS burst depending on respiratory burst oxidase homologs protein D (RBOHD) in Arabidopsis [17, 33]. The over-expression of homologous genes of SOS3 (Cluster-8107.8609, L2fc=1.700), OST1 (Cluster-8107.16572, L2fc=2.942), RBOHD (Cluster-8107.23569, L2fc=1.905), and RBOHF (Cluster-8107.2579, L2fc=3.997) suggested that Ca2+ signal sensors, ROS, CDPKs, and ABA were part of the regulatory network after silkworm feeding in mulberry (Figure 5A, Table 1). Since ROS-dependent Ca2+ influx and CDPKs can activate RBOHs, so they were likely to react closely with Ca2+ in mulberry early defense signaling. Herbivory and wounding rapidly activate MAPKs in the plant’s immune system. In mulberry, we found that the homolo-gous gene of MPK20 (Cluster-8107.12324, L2fc=1.930) was up-regulated after 3 h of silk-worm feeding, indicating that this gene positively regulated resistance to the silkworm.”

Page 10, line 80. While some of the mulberry transcripts show homology with Arabidopsis glucosinolate biosynthetic genes, one cannot claim the presence of glucosinolates (or nicotin, in figure 5) in mulberry. Hereto, a metabolite analysis should be performed. Wat can be stated is that these type of enzymes for which the differential expressed genes encode for (most likely CYP450 enzymes, but this is not clear form the presented data) are involved in specialized metabolism. Then the authors can refer to analogues in Arabidopsis, but be precise in formulation and claiming about specific metabolites present in mulberry.

Sorry for the confusion and thanks for suggestion. As suggestion, mulberry does not produce glucosinolates. We checked the manuscripts carefully and deleted all the sentences contained “glucosinolates” or “GS”. Besides, we added a new paragraph to dissed structural traits such as leaf surface wax, cell wall thickness and lignification form the first physical barrier to feeding by the silkworms.

Line 236-245

“MapMan enrichment analysis results show that, in mulberry, structural traits such as leaf surface wax, cell wall thickness and lignification form the first physical barrier to feeding by the silkworms. In plants, the physical properties of the wax layer as well as its chemical composition are important factors of preformed resistance. In the present study, three wax metabolism genes were up-regulated after 3 h of silkworm feeding, including wax synthase family protein coding gene (Cluster-4656.0, L2fc=4.483). In mulberry, the overexpression of cell wall precursor synthesis, cellulose and lignin genes after silkworm feeding (Figure 5 B-C), indicated the biosynthesis of lignin and cellulose and accelerates covalent linkage between lignin and other cell wall polymers, which is linked to the cell wall strengthening after silkworm feeding.”

Figure 6 is a nice figure that shows the fast and transient induction of the a-linolenic and mevalonate pathways in response to silkworm feeding. Please check carefully correct typing (e.g. ios-jasmonate should be isoleucine jasmonate / diteronoid mts be diterpenoid etc) throughout the figure. In panel B it seems that plastidical GGPP (diterpenoids and carotenoids) pathway are mixed with mevalonate FPP pathway. Are the authors aware of compartmentalisation of MVA and MEP pathways? The presented gene transcripts indicate mostly the cytosolic MVA pathway to be affected?

Sorry for the confusion and thanks for suggestion. We checked and modified text in the figure, “ios-jasmonate” was changed to “isoleucine jasmonate”, “diteronoid” was changed to “diterpenoid”. We speculated that the MVA pathway not MEP pathway was likely to be the main terpenoid pathway in response to silkworm infestation in mulberry.

Line 348-358

“In the present work, we showed that silkworm infestation up-regulated many genes at the mRNA level in mevalonate pathway (MVA), such as ACAT (Cluster-8107.10778, L2fc=3.619), HMGCS (Cluster-8107.12939, L2fc=2.634), HMGCR (Cluster-8107.15919, L2fc=4.3524), MVK (Cluster-8107.12465, L2fc=3.146), and FDPS (Cluster-8107.19051, L2fc=3.353) (Figure 6 B). However, essential genes of the non-mevalonate/MEP pathway were not dysregulated in mulberry damaged by the silkworm. Key genes in the mevalo-nate pathway were up-regulated. In cotton, the mevalonate pathway functions as a critical phytoalexin and provides constitutive and inducible resistance against various pests and diseases [24, 42]. We speculated that the MVA pathway not MEP pathway was likely to be the main terpenoid pathway in response to silkworm infestation in mulberry.”

I do not know whether mulberry contains indole diterpene alkaloids as is suggested in figure 6, but if so, the authors should include a reference. Otherwise, it is a similar issue as glucosinolates and nicotine as discussed before.

Sorry for the confusion and thanks for suggestion. We checked and modified text in the figure, “indole diterpene alkaloids” was deleted.

Line 141 & 149 – GS should be specialized metabolites.

Sorry for the confusion and thanks for suggestion. As suggestion, mulberry does not produce glucosinolates. We checked the manuscripts carefully and deleted all the sentences contained “glucosinolates” or “GS”.

Material and methods

Line 183 – reared on mulberry leaves or other??

Sorry for the confusion and thanks for suggestion. We modified this.

Line 375-376

“The silkworms were reared by mulberry leaves at the same institute under the same conditions.”

Line 200 – susceptible and resistant mulberry seedlings – this has not been addressed in the results and discussion section!

Sorry for the confusion and thanks for suggestion. We mistakenly wrote the description of seedlings as “susceptible and resistant”, and the seedlings we used were ordinary wild type. We modified this in the revised manuscript,

  1. 368-369

“Mulberry plants (M. alba) wildtype were cultivated at the Key Lab of Plant Stress Re-search, Shandong Normal University, Shandong Province, China.”

Line 237 – 12 independent libraries – this number does not align with the presented data in this manuscript

Sorry for the confusion and thanks for suggestion. We mistakenly wrote this. We modified it in the revised version.

  1. 419-420

A total of 9 independent transcript libraries were created for mulberry control/insect feeding leaves.

Conclusions

Line 271 – volatile (nor endogenous metabolites) analysis is not presented in this study

We mistakenly wrote “volatile analysis”. We checked the manuscripts carefully and deleted them.

Line 280 – again the glucosinolates issue... if the authors think this is actually a important issue, i suggest to include a metabolite analysis (or at least a solid reference) to demonstrate the presence of glucosinolates in mulberry.

Sorry for the confusion and thanks for suggestion. As suggestion, mulberry does not produce glucosinolates. We checked the manuscripts carefully and deleted all the sentences contained “glucosinolates” or “GS”.

At this stage of the writing process, i did not check the reference list nor the supplemental data.

Sorry for the confusion and thanks for suggestion. We have checked the reference list and the supplemental data in the revised version.
